# Acute Kidney Injury in Children with Polyuria: A Systematic Review

**DOI:** 10.3390/jcm15010351

**Published:** 2026-01-02

**Authors:** Giulio Rivetti, Mariantonia Braile, Anna Di Sessa, Paola Tirelli, Stefano Guarino, Emanuele Miraglia del Giudice, Isabella Guzzo, Pierluigi Marzuillo

**Affiliations:** 1Department of Woman, Child and of General and Specialized Surgery, Università Degli Studi Della Campania “Luigi Vanvitelli”, Via Luigi De Crecchio 2, 80138 Naples, Italy; giuliorivetti94@gmail.com (G.R.); brailemariantonia@gmail.com (M.B.); anna.disessa@unicampania.it (A.D.S.); paola.tirelli@unicampania.it (P.T.); stefano.guarino@policliniconapoli.it (S.G.); emanuele.miraglia@unicampania.it (E.M.d.G.); 2Division of Nephrology, Dialysis and Transplant Unit, IRCCS Bambino Gesù Children’s Hospital, Piazza Sant’Onofrio 4, 00165 Rome, Italy; isabella.guzzo@opbg.net

**Keywords:** acute kidney injury, AKI, children, diagnosis, pediatric, polyuria

## Abstract

**Objectives**: Children may present with acute kidney injury (AKI) and polyuria under certain conditions which can complicate diagnosis and management. This review seeks to synthesize AKI presentations in children with polyuria. **Methods**: Publications for this systematic review were searched using Embase, PubMed, and Scopus. Methodological quality assessment of the included study and case reports was performed. **Results**: From the selected studies, we obtained data on 32 patients with a mean age of 11.02 ± 2.82 years, including 13 males and 19 females. Among them, 26 presented with polyuria: 19 with diabetic ketoacidosis (DKA), 5 with new-onset type 1 diabetes mellitus (T1DM) without DKA, 1 with Bartter syndrome, and 1 with neuroblastoma. In 12 patients with DKA, data to calculate AKI prevalence were not available. Among the remaining 20 patients (all with polyuria), 9 (45%) developed AKI. AKI stage 3 was observed in 4 patients and stage 2 in 1 patient. For the remaining 5 patients with AKI, no information about the AKI stage was available. **Conclusions**: AKI can present with polyuria as part of its pathophysiological mechanism. A relationship between polyuria and AKI (with KDIGO stage ≥ 2) was found in metabolic disorders (DKA), nephrological diseases (Bartter syndrome), and oncological conditions (neuroblastoma).

## 1. Introduction

Acute kidney injury (AKI) is characterized by a sudden decline in kidney function [1]. It is a common condition, with recent studies reporting it complicates 1% to 25% of intensive care unit admissions and affects approximately 1% to 7% of all hospitalized patients [2,3,4].

Causes of AKI are broadly classified into three categories: pre-renal, intrinsic renal, and post-renal [1]. Pre-renal AKI results from reduced kidney perfusion, often caused by conditions such as hypovolemia that impair blood flow to the kidneys [5]. Common scenarios leading to pre-renal AKI in children include gastrointestinal losses due to vomiting or diarrhea [6], heart failure [7], acute respiratory infections [8,9], and sepsis [10]. Intrinsic renal AKI involves direct damage to the renal parenchyma. A common cause in clinical practice is acute tubular necrosis, often triggered by ischemic injury or exposure to nephrotoxins, such as certain medications and contrast agents [11]. Other intrinsic causes include glomerulonephritis and acute interstitial nephritis [11]. Finally, post-renal AKI occurs due to urinary tract obstruction, which increases pressure within the kidneys and causes subsequent injury [11].

Prompt recognition of AKI is paramount in improving overall patient outcomes. The Kidney Disease: Improving Global Outcomes (KDIGO) criteria are recommended for the diagnosis and staging of AKI, defining it based on an increase in serum creatinine and/or a reduction in urine output [12,13]. Urine output assessment has been advocated as one of the cheapest and most impactful biomarkers in AKI, emphasizing its practical value at the bedside [14]. Nevertheless, in clinical practice, oliguria—commonly defined as urine output < 0.5 mL/kg/h for at least 6 h—is often an early and important indicator of AKI and should always prompt careful evaluation and close monitoring [12].

While these parameters serve as widely accepted benchmarks, they have some limitations, particularly in conditions that predispose to AKI through uncommon mechanisms, such as cases presenting with polyuria.

Polyuria, defined as an increase in urine output exceeding normal thresholds, may indeed pose a diagnostic challenge. In fact, in certain conditions, patients with AKI may paradoxically present with an increased urine output due to a variety of pathophysiological mechanisms. For example, tubular epithelial cell injury may compromise the kidney’s concentrating ability, prompting an inadequate reabsorption of water and solutes. Additionally, impaired handling of nitrogenous wastes and other osmoles drives solute diuresis [15]. Consequently, despite overall renal dysfunction, patients with AKI may paradoxically exhibit high urine output [16]. This review seeks to synthesize AKI presentations in children with polyuria. 

### But How Can Polyuria Cause AKI?

Polyuria may accompany AKI and may also contribute to its development or progression, depending on the clinical context.

Osmotic diuresis (e.g., glycosuria in DKA, or high solute excretion) increases urinary water losses and can precipitate effective circulating volume depletion, thereby reducing renal perfusion and promoting pre-renal AKI and, if persistent, ischemic tubular injury [17,18].In intrinsic renal injury, particularly after ischemic or toxic insults, tubular epithelial damage can impair solute reabsorption and the generation/maintenance of the corticomedullary osmotic gradient, leading to a reduced response to vasopressin and an impaired urinary concentrating ability [13,15,19]. This may result in non-oliguric or polyuric AKI, especially during the recovery phase of AKI, where urine output may be preserved despite a reduced glomerular filtration rate.Polyuria itself can aggravate kidney injury by perpetuating negative fluid balance, hypernatremia, and electrolyte derangements (e.g., hypokalemia, hypophosphatemia), which may further compromise renal hemodynamics and tubular function [5,20,21].

Therefore, polyuria should not be interpreted as reassuring in children at risk of AKI; rather, it may represent either a marker of tubular dysfunction or a driver of hemodynamic stress, and it requires structured monitoring and management.

## 2. Materials and Methods

### 2.1. Information Sources and Search Strategy

A systematic review was performed to identify all studies reporting the association between AKI in pediatric patients and polyuria. This review was registered on PROSPERO with ID: CRD420250627229. The studies were searched in three different databases, Embase, Pubmed, and Scopus. The following keywords were mixed in different combinations for each data bank: “AKI” or “acute kidney injury”, and “polyuria” and “children”. Boolean operators “AND” and “OR” were employed in our search plan. The research in databases ended on 8 December 2024. To assess the eligibility of each article, each PICO (Population, Intervention, Comparison, Outcome) [22] element was identified as outlined below. Population (P): Pediatric patients diagnosed with AKI. Intervention (I): AKI presentation in children with polyuria. Comparison (C): The comparison of patients diagnosed with AKI, with and without polyuria. Outcome (O): The impact of polyuria on AKI and its associated clinical outcomes. The various topics analyzed in this systematic review are summarized in a checklist according to the Preferred Reporting Items for Systematic Reviews and Meta-Analyses (PRISMA) guidelines 2020 Statement [23]. PRISMA checklist is reported in the Appendix A.

### 2.2. Eligibility Criteria

One author searched the combination of keywords using the Boolean operators AND and OR across the three different databases. Afterwards, the same author combined the three spreadsheets into a single spreadsheet, deleting duplicates. Then, independently, two authors screened each remaining article based on the following inclusion criteria: (i) studies focused on children affected by AKI; (ii) studies based on measurement of polyuria; (iii) studies on humans. The articles were evaluated as being not eligible considering the following specific exclusion criteria: (i) articles missing one or more keywords; (ii) conference abstract; (iii) editorial, or letter, or book; (iv) irrelevant articles to the main subject; (v) no English language; (vi) and review (narrative, systematic or meta-analysis); (vii) non-human subjects (e.g., animals). The reason for exclusion of each record is reported in Appendix A. The title and abstract of all studies found in the search were independently examined by two reviewers who applied the eligibility criteria. In case of disagreement between the reviewers, a third reviewer was consulted.

### 2.3. Data Extraction and Quality Process

Five manuscripts were selected for the final review after the screening procedure. Following selection, data extraction was carried out by manual curation. Two authors extracted the data and then separately described the findings of each paper.

The “Newcastle–Ottawa Scale” (NOS) assessment instrument was used to evaluate the case-control studies’ quality [24]. Eight components constitute the structure of the case-control research form, which is separated into three areas: exposure, comparability, and selection. Each aspect has a score that ranges from 0 and 9 points. A better study quality correlates with a higher score. Three questions pertaining to relevant articles were used to determine each study’s risk of bias: (i) bias pertaining to the clinical data of patients who were enrolled; (ii) bias pertaining to measurement of polyuria; (iii) bias pertaining to exposure determination. Every article was independently reviewed by three authors. Each component was classified as having a “low risk of bias” if its score was greater than 7; if not, it was classified as having a “high risk of bias” [24].

Next, the “Case Report Guidelines” (CARE) were used to assess whether authors adhered to them in their case reports to improve their quality [25]. These guidelines consist of a 13-item checklist that emphasizes key components, including patient information, clinical findings, diagnostic assessments, and treatment interventions.

Finally, “A MeaSurement Tool to Assess Systematic Reviews 2” (AMSTAR 2), containing 16 criteria, was used to evaluate the methodological quality of this systematic review (Appendix A) [26]. Several crucial and non-critical topics are examined in the 16 components. Relevant fields of research include protocol registration, literature searches, study exclusion, potential bias, the applicability of meta-analytic approaches, and heterogeneity evaluation. Based on the existence of critical and non-critical flaws, the AMSTAR 2 tool divides systematic reviews into four confidence degrees (high, moderate, low, and critically low). Nevertheless, the tool does not provide a numerical score for the review.

## 3. Results

### 3.1. Literature Research

The flowchart provides a clear illustration of the process for literature searching (Figure 1).

A total of 344 articles were obtained by interrogating Pubmed, Embase, and Scopus based on the following keywords: “AKI” or “acute kidney injury”, and “polyuria” and “children”. After discarding duplicates, 301 articles were assessed according to the eligibility criteria. To complete the screening procedure, five articles were included in this systematic review in order to elucidate the impact of AKI on pediatric patients with polyuria.

### 3.2. Study Characteristics

The main characteristics of all the included studies are reported in Table 1.

All the included studies were published between 2013 and 2023 and were conducted on human subjects. Furthermore, one study was retrospective [27], and four were case reports [28,29,30,31]. The number of subjects enrolled in the studies varied from 1 to 19, with the majority being females. All the studies selected were conducted on polyuric patients. Urine and blood tests were performed in all studies.

### 3.3. Clinical Outcome

From the selected studies, we obtained data on 32 patients with a mean age of 11.02 ± 2.82 years, including 13 males and 19 females. Among them, 26 presented with polyuria: 19 with diabetic ketoacidosis (DKA), 5 with new-onset T1DM without DKA, 1 with Bartter syndrome, and 1 with neuroblastoma. In 12 patients with DKA, data to calculate AKI prevalence were not available [27]. Therefore, we considered a final population of 20 patients (all with polyuria). Nine out of 20 patients (45%) developed AKI. AKI stage 3 was observed in four patients, stage 2 in one patient. No information about the AKI stage of the remaining five patients were available [28,29,30,31].

The studies analyzed in this systematic review indicate a relationship between polyuria and AKI across three major categories of conditions: metabolic disorders, nephrological diseases, and oncological conditions.

### 3.4. Metabolic Disorders

In the study by Han et al. [27] the authors reported a concerning increase in the incidence of pediatric DKA following the onset of the COVID-19 pandemic. A key finding was the predominance of polyuria as a symptom: among the 19 patients with DKA, 13 (68.4%) exhibited polyuria. However, the participants’ urine output was not reported and the cut-off to define polyuria was not indicated. Notably, the prevalence of polyuria was significantly higher in the post-COVID group (100%) compared to the pre-COVID group (40%, 50%, and 66.7% across different years). Moreover, eight patients (42.1%) presented with AKI. No data about prevalence of polyuria in patients developing AKI in the pre-COVID group are available, while separately evaluating the post-COVID group, the prevalence of AKI in polyuric patients was 57.1% (4/7). In the study conducted by John et al. [31] all the eight patients (100%) evaluated presented with polyuria (100%) and one progressed to AKI (12.5%). Also in this study, the participants’ urine output was not reported and the cut-off to define polyuria was not indicated. In the case report by Tinti et al. [29], three patients with DKA were described. All these developed AKI. Two patients experienced AKI with polyuria (cut-off for polyuria not specified and amount of diuresis not indicated) and were diagnosed based on an increased creatinine value (KDIGO criteria) [12]. The other patient initially presented with polyuria and subsequently developed AKI with oliguria [31]. In detail, patient 1 (13 years old, female) presented a creatinine value of 1.5 mg/dL (KDIGO stage 2), patient 2 (10 years old, male) presented a creatinine value not directly mentioned but described as increased by 4 times the baseline (KDIGO stage 3 AKI) while patient 3 (11 years old, female) presented at admission a creatine value of 0.69 mg/dL (estimated glomerular filtration rate = 75 mL/min/1.73 m^2^) and oliguria (AKI pRIFLE stage Injury, KDIGO stage 2 [32]).

### 3.5. Nephrological Diseases

In the case report by Yang et al. [24], the authors detail the case of an extremely preterm infant diagnosed with transient antenatal Bartter syndrome. The male infant was observed soon after the delivery had increased urinary output (5.0 mL/kg per hour on average), accompanied by electrolyte imbalances (hypokalemia and metabolic alkalosis). Nevertheless, soon after birth, the patients presented a transient hematuria and serum creatinine level of 1.31 mg/dL at the 6th day of life (KDIGO neonatal AKI stage 3 [33]).

### 3.6. Oncological Conditions

In the case report by Poggi et al. [28], the authors describe a 10-month-old female child diagnosed with neuroblastoma who presented with AKI (KDIGO Stage 3), hyponatremic-hypertensive-like syndrome, and nephrotic proteinuria. Polyuria (diuresis 125 mL/kg/24 h) was a prominent symptom in this patient and was attributed to both glomerular hyperfiltration and tubulointerstitial damage. The excessive urine output resulted in severe dehydration, hypovolemia, and hemoconcentration, which compounded the renal dysfunction.

### 3.7. Quality Assessment and Risk of Bias Across Studies

The results from the NOS assessment of the selected studies provide valuable insights into their methodological quality and strengths [24]. Only one article could be evaluated with this approach being a retrospective case-control study. Han et al. [27] received moderate scores of 6 stars, suggesting that although they employed clear case definitions, their control selection and representativeness could be improved to enhance the robustness of their findings (Table 2).

NOS Selection:
Case Definition: Assesses whether a clear definition of cases was provided and documented.Representativeness of Cases: Evaluates if the population studied is representative of the general population with similar conditions.Selection of Controls: Determines how well controls are selected (if applicable) to make comparisons.Definition of Controls: Checks if the control group’s characteristics are well defined.
NOS Comparability:
Comparability of Cases and Controls: Examines if there are adjustments made for confounding factors that could affect study outcomes.NOS Exposure:
Ascertainment of Exposure: Looks at how reliably the exposure (in this case, outcomes or conditions relevant to the studies) was measured.Same Method of Ascertainment for Cases and Controls: Assesses if the same methods of data collection were used for both cases and controls to ensure consistency.Non-Response Rate: Evaluates the response rates between groups, considering characteristics of non-responders if applicable.

Overall, these evaluations underscore the need for ongoing efforts to improve the methodological rigor of studies in pediatric health, particularly around critical conditions like diabetic ketoacidosis, to allow for more definitive conclusions and more effective interventions. The application of the CARE guidelines to the four included case reports [28,29,30,31] demonstrates a commitment to quality and thoroughness in clinical documentation [19] (Table 3).

In the report by Poggi et al. [28], which details a 10-month-old child with neuroblastoma, the title is descriptive, and the authors provide comprehensive clinical details, including diagnostic findings and treatment interventions, along with follow-up outcomes. Tinti et al. [29] present a detailed report on three children with severe acidosis but low ketones at T1D onset, highlighting the diagnostic challenge of differentiating acidosis caused by DKA from that due to AKI. They emphasize the importance of measuring ketones, as high ketone levels indicate acidosis due to DKA, while low ketone levels suggest acidosis due to AKI.

Meanwhile, Yang et al. [30] focus on a case of transient antenatal Bartter syndrome in an extremely preterm infant, offering detailed information on diagnosis, management, and clinical progress. Similarly, the case series “Clinical Profile of Childhood Type 1 in Jos, Nigeria” adheres to the CARE guidelines by providing a structured overview of patient demographics, clinical findings, diagnostic assessments, and treatment interventions. This adds to the depth of understanding regarding the management of childhood diabetes in that region and enhances the clarity and reliability of the case reports [28,29,30,31].

While all four reports generally comply with the CARE guidelines, emphasizing follow-up data, broader clinical implications in each case could have further enhanced their contributions to the existing literature. By addressing these aspects, authors may have provided a more comprehensive perspective on the long-term management and outcomes of the conditions they report.

## 4. Discussion

Following PRISMA 2020 standards, this systematic review aimed to clarify AKI presentation that includes polyuria. Although oliguria is a key diagnostic criterion for AKI according to the KDIGO guidelines [12], we found that in a subset of pediatric patients AKI could be associated with polyuria rather than oliguria. Among the 20 patients with polyuria included in our final analysis, nine (45%) developed AKI, with severity ranging from KDIGO stage 2 to stage 3.

The study by Han et al. [27] indicated that 42.1% of patients with DKA presented with AKI, while separately evaluating the post-COVID group, the prevalence of AKI in polyuric patients was 57.1%. However, the study faces several criticisms and weaknesses. First, its retrospective nature may introduce biases related to data collection, as the accuracy of medical records and patient recall can vary. Additionally, the study employs a two-center design, which may limit the generalizability of the findings to broader populations or regions, as patient demographics and healthcare practices can differ significantly between institutions. Moreover, the reliance on a relatively small sample size (19 patients) may affect the strength of the conclusions and the statistical analyses.

In the study by John et al. [31], all eight patients evaluated exhibited polyuria. Additionally, one patient progressed to AKI.

In the case report by Tinti et al. [29], the authors emphasize the critical role of ketone monitoring in a pediatric patient at T1DM onset with AKI, highlighting that low ketone levels despite severe acidosis may indicate an alternative etiology, such as renal tubular dysfunction or lactic acidosis, likely due to impaired bicarbonate reabsorption and metabolic stress, thus guiding appropriate management and avoiding unnecessary DKA treatment protocols. Distinction is critical, as metabolic acidosis may result from impaired renal bicarbonate reabsorption and altered acid–base homeostasis due to AKI, rather than excessive ketoacid production. All three patients presented with polyuria at T1DM onset and developed AKI; however, one patient later developed oliguria and met the oliguria-based criterion (Stage 2 KDIGO) [33], which reflected a milder AKI compared with the other two, who progressed to Stage 3 KDIGO [12] based solely on elevated serum creatinine.

None of the patients experienced stage 1 AKI. These data suggest that polyuria could be associated with more severe forms of AKI, although a definitive association cannot be established. Taken together, in DKA cases, polyuria is a hallmark symptom driven by osmotic diuresis secondary to hyperglycemia [17,18,34,35]. Due to large urinary output, these patients often develop severe dehydration, hypovolemia, and renal hypoperfusion, culminating in AKI. Therefore, the reviewed studies show cases of patients with polyuria developing AKI without oliguria in DKA, suggesting the importance of recognizing polyuria as a precursor to renal dysfunction rather than simply as clinical manifestation of the onset of T1DM.

Yang et al. [30] emphasized the need for careful evaluation of polyuria in neonates and infants, as it may signal serious underlying renal disorders requiring prompt management. For this article, one major concern is the report’s focus on a single case, which limits the generalizability of the findings to a broader population. Congenital tubulopathies such as Bartter syndrome, however, frequently lead to chronic polyuria due to impaired tubular reabsorption, which predisposes patients to volume depletion and prerenal AKI [36]. Other tubulopathies associated with significant polyuria, such as Gitelman syndrome, nephrogenic diabetes insipidus, and Fanconi syndrome, can contribute to the development of AKI under specific circumstances [37,38]. These findings underscore the clinical relevance of polyuria in driving kidney injury, particularly when fluid and electrolyte imbalances are not adequately managed. In these cases, polyuria is not a protective mechanism but a pathological feature that directly contributes to renal compromise.

The case report described by Poggi et al. [28] shows the critical role of polyuria as both a symptom and a driver of renal injury in oncological diseases like neuroblastoma. However, this is only a case report, and no further generalizations can be obtained. However, oncological conditions like neuroblastoma can illustrate the multifaceted nature of polyuria in AKI. Here, mechanisms such as glomerular hyperfiltration, tubular damage, and paraneoplastic effects converge, leading to significant renal dysfunction despite the presence of elevated urinary output [28].

The consistent association between AKI and polyuria across diverse clinical contexts suggests that a broader understanding of urinary patterns in AKI is necessary. Relying solely on oliguria as a primary diagnostic criterion may delay recognition and treatment in patients presenting with polyuria.

The main limitation of this systematic review is the limited number of studies investigating the relationship between AKI and polyuria. This also made it impossible to compare the characteristics of AKI in patients with and without polyuria, as originally planned when defining the PICO before conducting the literature search. A further limitation is the heterogeneity in how polyuria was defined across the included studies, with variable criteria and, in several cases, the absence of explicit quantitative urine output thresholds, thereby complicating data synthesis and comparisons. Future studies should adopt a standardized definition of polyuria to improve reproducibility and comparability (e.g., urine output > 4 mL/kg/h in children and >2 L/m^2^/day in adolescents) [21].

Furthermore, the only observational study included (Han et al., 2021, [27]) achieved a moderate NOS score (6/9), reflecting potential risks of bias such as incomplete adjustment for confounding, selection bias, and limitations inherent to retrospective data collection. Moreover, the remaining included publications were case reports which are valuable for hypothesis generation and for highlighting atypical clinical presentations but also intrinsically subject to publication and reporting bias and tend to preferentially describe unusual or severe cases, not allowing a real estimation of incidence/prevalence or robust assessment of associations.

Despite these limitations, this systematic review may pave the way for future multicenter studies aimed at describing the clinical characteristics of AKI in patients with and without polyuria and at assessing the impact of polyuria on AKI development and severity.

## 5. Conclusions

In conclusion, AKI may present with polyuria. In several pediatric settings—such as DKA, selected tubulopathies, and neuroblastoma—polyuria can contribute to AKI progression by promoting or aggravating volume depletion and/or solute diuresis; therefore, careful fluid and electrolyte management is crucial in polyuric children to prevent worsening kidney injury. In children in whom polyuria is a key determinant of AKI risk (or AKI worsening), hydration should not follow “standard” maintenance approaches alone (e.g., Holliday–Segar) but should be dynamically tailored to measured urine output and ongoing losses. However, polyuria is not always the pathophysiological driver of AKI: intrinsic renal processes (e.g., drug-induced acute interstitial nephritis) may cause AKI even in patients who remain normovolemic. In these cases, timely recognition and elimination of the causal factor (including prompt withdrawal of the offending medication) and consideration of the disease-specific therapy when indicated are essential to improve outcomes.

## Figures and Tables

**Figure 1 jcm-15-00351-f001:**
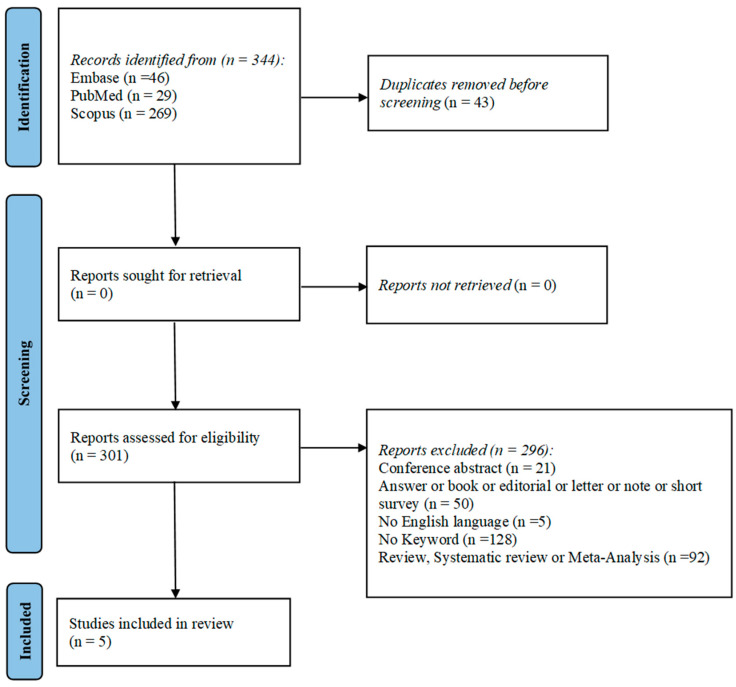
PRISMA flow-diagram showing research strategy.

**Table 1 jcm-15-00351-t001:** Characteristics and outcome of the studies of patients assessed for eligibility criteria.

Author, Year	Study Design	Enrolled Population, *n*	Measurement of Specific Biomarkers	Outcome
Han et al. (2021) [27]	Retrospective Study	19 in total: Pre-COVID-19 Group (12 patients): This group included patients who were diagnosed with DKA before the COVID-19 pandemic began Post-COVID-19 Group (7 patients): This group consisted of patients diagnosed with DKA after the onset of the COVID-19 pandemic	Blood Glucose Levels Ketone Bodies Electrolyte Levels Anion Gap ABG	Out of 19 DKA patients, 8 presented AKI (42.1%). Particularly in the Post-COVID Group 4 patients presented AKI and polyuria (57.1%)
Poggi et al. (2021) [28]	Case Report	1 child affected by neuroblastoma	Serum Electrolytes Serum Creatinine Urinalysis Protein Levels BUN	Presentation of AKI and polyuria in a hyponatremic-hypertensive syndrome
Tinti et al. (2023) [29]	Case Report	3 children affected by T1DM	Serum Ketones Blood Glucose Levels Electrolyte Levels BUN and Serum Creatinine ABG	All the patients described presented with AKI, 2/3 with polyuria and 1/3 with oliguria.
Yang et al. (2022) [30]	Case Report	1 extremely preterm infant affected by Bartter syndrome	Serum Electrolytes Urine Electrolytes BUN and Creatinine Calcium Levels	Transient antenatal Bartter syndrome diagnosed with polyuria and AKI
John et al. (2013) [31]	Case Report	8 children affected by type 1 diabetes mellitus	Serum Ketones Blood Glucose Levels Electrolyte Levels BUN and Serum Creatinine ABG	One patient out of 8 enrolled presented both AKI and polyuria (12.5%)

Abbreviation: ABG, arterial blood gases; AKI, acute kidney injury; BUN, blood urea nitrogen; DKA, diabetic ketoacidosis, T1DM, Type 1 diabetes mellitus.

**Table 2 jcm-15-00351-t002:** NOS for case-control study quality assessment.

Authors	NOS Selection (Up to 4 Stars)	NOS Comparability (Up to 2 Stars)	NOS Exposure (Up to 3 Stars)	NOS Score (Max. 9 Stars)
Han et al. (2021) [27]	★ ★	★★	★ ★	6

**Table 3 jcm-15-00351-t003:** CARE Guideline for case report study quality assessment.

CARE Guideline Item	Poggi et al. (2021) [28]	Tinti et al. (2023) [29]	Yang et al. (2022) [30]	John et al. (2013) [31]
Title	Descriptive, includes condition and patient information	Clearly indicates focus on ketone monitoring and context	Accurately describes case and genetics	Clearly reflects the content, specifying “Clinical Profile of Childhood Type 1 Diabetes”
Abstract	Includes a summary of the presentation, findings, and relevance of the case	Provides an abstract summarizing the reason for the case report and key findings	Abstract highlights the significance of the presented case and its implications	Abstract summarizes demographic and clinical aspects of childhood type 1 diabetes in Jos, Nigeria
Keywords	Well defined	Well defined	Well defined	Well defined
Patient Information	Detailed age and relevant medical history	Patient details including age and medical history	Comprehensive information on birth history and antenatal complications	Comprehensive information on demographic details
Clinical Findings	Thorough description of symptoms and clinical examinations	Detailed observations about symptoms, signs, and lab results	Thorough description of symptoms and clinical status	Detailed observations about symptoms, signs, and lab results.
Timeline	Well-defined, describing presentation of AKI and associated symptoms, identification of neuroblastoma via imaging; documenting renal function changes, fluctuations in hypertension, and scheduling follow-up	Well-defined, describing development of DKA signs	Well-defined, describing signs and symptoms prenatally to at the time of birth	Well-defined. Consistent monitoring for complications in type 1 diabetes such as nephropathy initiated, including regular follow-up
Diagnostic Focus and Assessment	Discusses imaging studies and lab tests for neuroblastoma	Criteria for assessing DKA severity and renal impairment	Methods and criteria for diagnosing Bartter syndrome, including genetic testing	Details the methods used for diagnosing type 1 diabetes, including laboratory tests.
Therapeutic Intervention	Treatment strategies, including surgical intervention and supportive care	Specific treatment and management strategies, emphasizing ketone monitoring	Management approaches taken for Bartter syndrome	Specific treatment and management strategies, emphasizing insulin therapy
Follow-Up and Outcomes	Information on follow-up care and prognosis	Subsequent care and patient’s recovery trajectory	Monitoring and clinical progression post-intervention	Tracking and clinical development after the intervention
Discussion	Analyses findings in context of existing literature	Highlights importance of ketone monitoring within existing research	Analyses implications and contributions to understanding Bartter syndrome	Considers the implications for practice and research while analyzing the findings in light of the body of existing literature
Patient Perspective	Does not specify patient or caregiver perspectives; focuses primarily on clinical data	May include caregiver insights into their child’s condition and treatment adherence, but not thoroughly documented	Comments related to the family’s experience can be inferred but are not explicitly captured	Patient perspectives may not be expressly detailed, focusing instead on presenting clinical data
Informed Consent	Not defined	Well-defined	Well-defined	Not defined

Abbreviation: AKI, acute kidney injury; DKA, diabetic ketoacidosis.

## Data Availability

Uploaded as Appendix A.

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
