# Peer review of "Acute Kidney Injury in Children with Polyuria: A Systematic Review"

_jcm, 2026, doi:10.3390/jcm15010351_

Round 1
Reviewer 1 Report
Comments and Suggestions for Authors
The introduction should be improved. Despite the many limitations from the review presented and described by the authors this review draws attention to an important point, which is the possibility of polyuric AKI. Nevertheless, oliguria is an important clue to the possibility of AKI and it should be highlighted. I suggest adding the reference: Goldstein, Stuart. (2020). Urine Output Assessment in Acute Kidney Injury: The Cheapest and Most Impactful Biomarker. Frontiers in Pediatrics. 7. 565. 10.3389/fped.. 2019.00565
Also the thresholds that define oliguria should be included.
It's important to add in the conclusion that polyuria can result in acute renal failure (ARF), but this is not always the pathophysiological mechanism. In cases of interstitial nephritis, for example, even if the patient remains normovolemic, AKI can occur due to interstitial changes. If the causal factor is not eliminated from the diagnosis, discontinuation of the drug, or even corticosteroid therapy, the outcome can be devastating.
Author Response
The introduction should be improved. Despite the many limitations from the review presented and described by the authors this review draws attention to an important point, which is the possibility of polyuric AKI. Nevertheless, oliguria is an important clue to the possibility of AKI and it should be highlighted. I suggest adding the reference: Goldstein, Stuart. (2020). Urine Output Assessment in Acute Kidney Injury: The Cheapest and Most Impactful Biomarker. Frontiers in Pediatrics. 7. 565. 10.3389/fped.. 2019.00565
Answer: thank you for your comment, we improved the introduction as recommended (please see paragraph “Introduction”). We also highlighted that AKI, especially in children can be related to oliguria and added the reference as suggested (please see line 52-56)
Also the thresholds that define oliguria should be included.
Answer: we added the definition of oliguria (as a reduction of urinary output <0,5ml/kg/h for at least 6 hours) in lines 54-56.
It's important to add in the conclusion that polyuria can result in acute renal failure (ARF), but this is not always the pathophysiological mechanism. In cases of interstitial nephritis, for example, even if the patient remains normovolemic, AKI can occur due to interstitial changes. If the causal factor is not eliminated from the diagnosis, discontinuation of the drug, or even corticosteroid therapy, the outcome can be devastating.
Answer: we specified that in patients with AKI and polyuria this latter it’s not always the main pathophysiological mechanism since in some conditions that are accompanied by polyuria such as tubulointerstitial nephritis, AKI can be still present in normovolemic patients since the renal damage is perpetued by the interstitial changes (please see lines 363-371).
Reviewer 2 Report
Comments and Suggestions for Authors
This manuscript presents a systematic review addressing an important and understudied topic: the association between polyuria and AKI in children. The relevance of the work is high, as polyuria is traditionally associated with other conditions (e.g., diabetes insipidus) and is rarely considered as a potential symptom or risk factor for AKI. The authors rightly note that the KDIGO clinical criteria are primarily oriented towards oliguria, which may lead to under-recognition of AKI in patients presenting with polyuria.
1. Only five studies were included (one retrospective study and four case reports). This is a significant limitation, which the authors acknowledge, but it reduces the statistical power and generalizability of the conclusions. The included studies used varying definitions of polyuria, often without specifying quantitative cut-offs for urine output. This complicates data synthesis and comparison.
Recommendation: For future research, a standardized definition of polyuria should be applied (e.g., >4 mL/kg/h in children, >2 L/m²/day in adolescents).
2. The sole retrospective study (Han et al., 2021) received a moderate NOS score (6/9). The remaining works are case reports, which limits the overall level of evidence.
Recommendation: The authors should discuss in more detail how the limitations of the primary studies affect the interpretation of the systematic review's findings.
3. The authors correctly note that polyuria can be both a cause and a consequence of AKI, but the pathophysiological mechanisms underlying this interplay are not sufficiently explored. The practical recommendation for careful hydration management in children with polyuria is important but would be strengthened by proposing more specific clinical management algorithms.
Recommendation: A paragraph discussing potential pathophysiological mechanisms of polyuria in AKI (e.g., osmotic diuresis, tubular damage, impaired concentrating ability) should be added to the Discussion section.
Conсlusion: The article holds scientific interest, but the number of studies included in the analysis is very limited. I recommend revising the search and data analysis strategies to incorporate a larger number of articles, as well as addressing the aforementioned comments.
Author Response
This manuscript presents a systematic review addressing an important and understudied topic: the association between polyuria and AKI in children. The relevance of the work is high, as polyuria is traditionally associated with other conditions (e.g., diabetes insipidus) and is rarely considered as a potential symptom or risk factor for AKI. The authors rightly note that the KDIGO clinical criteria are primarily oriented towards oliguria, which may lead to under-recognition of AKI in patients presenting with polyuria.
Only five studies were included (one retrospective study and four case reports). This is a significant limitation, which the authors acknowledge, but it reduces the statistical power and generalizability of the conclusions. The included studies used varying definitions of polyuria, often without specifying quantitative cut-offs for urine output. This complicates data synthesis and comparison. Recommendation: For future research, a standardized definition of polyuria should be applied (e.g., >4 mL/kg/h in children, >2 L/m²/day in adolescents).
Answer: We thank the Reviewer for this important comment. We agree that the inclusion of only five studies represents a major limitation of the paper. In addition, we acknowledge that the included studies adopted heterogeneous definitions of polyuria, often lacking a quantitative urine output cut-off, which further complicates data synthesis and cross-study comparisons. We have therefore expanded the Limitations section accordingly and added a clear recommendation for future research to apply a standardized definition of polyuria (e.g., >4 mL/kg/h in children and >2 L/m²/day in adolescents-please see lines 343-345).
- The sole retrospective study (Han et al., 2021) received a moderate NOS score (6/9). The remaining works are case reports, which limits the overall level of evidence.
Recommendation: The authors should discuss in more detail how the limitations of the primary studies affect the interpretation of the systematic review's findings.
Answer: We thank the Reviewer for this important comment. We agree that the overall level of evidence is limited, given that only one study was retrospective (Han et al., 2021; moderate NOS score 6/9) and the remaining reports were case reports. We have therefore expanded the limitations section to clarify how the methodological limitations of the primary studies may influence the interpretation of our findings, particularly in terms of risk of bias, limited generalizability, inability to infer causality, and heterogeneity in definitions and reporting (please see lines 346-353).
- The authors correctly note that polyuria can be both a cause and a consequence of AKI, but the pathophysiological mechanisms underlying this interplay are not sufficiently explored. The practical recommendation for careful hydration management in children with polyuria is important but would be strengthened by proposing more specific clinical management algorithms.
Recommendation: A paragraph discussing potential pathophysiological mechanisms of polyuria in AKI (e.g., osmotic diuresis, tubular damage, impaired concentrating ability) should be added to the Discussion section.
Answer: We thank the Reviewer for this constructive comment. We agree that the pathophysiological interplay between polyuria and AKI deserves deeper discussion. We have therefore expanded the Discussion by adding a dedicated paragraph outlining the main mechanisms through which polyuria may occur in AKI and/or contribute to AKI development (including osmotic diuresis, tubular injury with impaired concentrating ability, and solute-driven diuresis) (please see lines 70-89). In addition, we have strengthened the practical implications by clarifying that, in children in whom polyuria is a key determinant of AKI risk (or AKI worsening), hydration should not follow “standard” maintenance approaches alone (e.g., Holliday–Segar), but should be dynamically tailored to measured urine output and ongoing losses, with close electrolyte monitoring (please see lines 364–369 of the new version of the manuscript).
Conсlusion: The article holds scientific interest, but the number of studies included in the analysis is very limited. I recommend revising the search and data analysis strategies to incorporate a larger number of articles, as well as addressing the aforementioned comments.
Answer: we thank the Reviewer for the comment. We agree that the number of eligible studies is limited and we have highlighted this as a major limitation of the current evidence base. However, we maximized the sensitivity of our search strategy by using broad, non-restrictive keywords and combinations in advanced search fields (e.g., “acute kidney injury” OR “AKI” AND children AND polyuria), and by incorporating controlled vocabulary where applicable. Importantly, within MeSH, the term “Acute Kidney Injury” maps to multiple related entry terms and synonyms including :
- Kidney Injuries, Acute
- Kidney Injury, Acute
- Acute Renal Injury
- Acute Renal Injuries
- Renal Injuries, Acute
- Renal Injury, Acute
- Kidney Failure, Acute
- Acute Kidney Failures
- Kidney Failures, Acute
- Acute Kidney Failure
- Acute Renal Failure
- Acute Renal Failures
- Renal Failures, Acute
- Renal Failure, Acute
- Renal Insufficiency, Acute
- Acute Renal Insufficiencies
- Renal Insufficiencies, Acute
- Acute Kidney Insufficiency
- Acute Renal Insufficiency
- Kidney Insufficiency, Acute
- Acute Kidney Insufficiencies
- Kidney Insufficiencies, Acute
which were inherently captured by our strategy. Despite this comprehensive approach, only a small number of studies specifically addressing AKI in the context of polyuria in pediatric patients met the predefined inclusion criteria. In our view, therefore, the limited number of included articles primarily reflects a scarcity of focused primary research in this field rather than an overly restrictive search strategy.
Round 2
Reviewer 2 Report
Comments and Suggestions for Authors
All comments have been corrected by the authors. I recommend the article for publication.